# Candidate genetic variants and antidepressant-related fall risk in middle-aged and older adults

A. C. Pronk[1‡]*, L. J. Seppala[1‡], K. Trajanoska[2], N. Stringa[3], B. van de Loo[1,3], L. C. P. G. M. de Groot[4], N. M. van Schoor[3], F. Koskeridis[5], G. Markozannes[5], E. Ntzani[5,6,7], A. G. Uitterlinden[2], F. Rivadeneira[2], B. H. Stricker[8], N. van der Velde[1]

1 Department of Internal Medicine, Section of Geriatric Medicine, Amsterdam Public Health Research Institute, Amsterdam UMC, University of Amsterdam, Amsterdam, the Netherlands, 2 Department of Internal Medicine, Erasmus University Medical Center, Rotterdam, the Netherlands, 3 Department of Epidemiology and Biostatistics, Amsterdam UMC, Vrije Universiteit Amsterdam, Amsterdam Public Health Research Institute, Amsterdam, the Netherlands, 4 Department of Human Nutrition and Health, Wageningen University, Wageningen, the Netherlands, 5 Department of Hygiene and Epidemiology, School of Medicine, University of Ioannina, Ioannina, Greece, 6 Department of Health Services, Policy and Practice, Center for Research Synthesis in Health, School of Public Health, Brown University, Providence, RI, United States of America, 7 Center for Evidence Synthesis in Health, Brown University School of Public Health, Providence, RI, United States of America, 8 Department of Epidemiology, Erasmus University Medical Center, Rotterdam, the Netherlands

‡ ACP and LJS are shared first co-authors on this work.
* a.c.pronk@amsterdamumc.nl

## Abstract

### Background

Antidepressant use has been associated with increased fall risk. Antidepressant-related adverse drug reactions (e.g. orthostatic hypotension) depend partly on genetic variation. We hypothesized that candidate genetic polymorphisms are associated with fall risk in older antidepressant users.

### Methods

The association between antidepressant use and falls was cross-sectionally investigated in a cohort of Dutch older adults by logistic regression analyses. In case of significant interaction product term of antidepressant use and candidate polymorphism, the association between the variant genotype and fall risk was assessed within antidepressant users and the association between antidepressant use and fall risk was investigated stratified per genotype. Secondly, a look-up of the candidate genes was performed in an existing genome-wide association study on drug-related falls in antidepressant users within the UK Biobank. In antidepressant users, genetic associations for our candidate polymorphisms for fall history were investigated.

### Results

In antidepressant users(n = 566), for rs28371725 (*CYP2D6\*41*) fall risk was decreased in TC/variant allele carriers compared to CC/non-variant allele carriers (OR = 0.45, 95% CI

**Data Availability Statement:** Because of restrictions based on privacy regulations and informed consent of the participants, data cannot be made freely available in a public repository. Data

from the LASA database are available for use for specific research questions provided that an agreement is made up (https://www.lasavu.nl/data/availability_data/availability_data.htm). Rotterdam study data can be obtained upon request. Requests should be directed towards the management team of the Rotterdam Study (secretariat.epi@erasmusmc.nl), which has a protocol for approving data request. B-PROOF data can be obtained upon request. The access committee may be contacted by: Adfice-it@amsterdamumc.nl. Data from the UK Biobank are available on application to the UK Biobank Access Committee. The Access Committee may be contacted by email (access@ukbiobank.ac.uk) or through the UK Biobank web page (http://www.ukbiobank.ac.uk).

**Funding:** This work was supported by the Clementine Brigitta Maria Dalderup fund (project numbers 20713 and 7303, receiver N. van der Velde), which is an Amsterdam University fund (https://www.auf.nl/en/about-the-fund/about.html). The funder had no role in study design, data collection and analysis, decision to publish, or preparation of the manuscript. The initial B-PROOF study has received funding so far by the Netherlands Organization for Health Research and Development (ZonMw, Grant 6130.0031), The Hague; unrestricted grant from NZO (Dutch Dairy Association), Zoetermeer; Orthica, Almere; Netherlands Consortium Healthy Ageing (NCHA) Leiden/Rotterdam; Ministry of Economic Affairs, Agriculture and Innovation (project KB-15-004-003), The Hague; Wageningen University, Wageningen; VUmc, Amsterdam; Erasmus Medical Center, Rotterdam. The Longitudinal Aging Study Amsterdam (LASA) is largely supported by a grant from the Netherlands Ministry of Health, Welfare and Sports, Directorate of Long-Term Care. The data collection in 2012–2013 was financially supported by the Netherlands Organization for Scientific Research (NWO) in the framework of the project "New Cohorts of young old in the 21st century" (File Number 480-10-014). The Rotterdam Study is supported by the Erasmus MC University Medical Center and Erasmus University Rotterdam; The Netherlands Organisation for Scientific Research (NWO); The Netherlands Organisation for Health Research and Development (ZonMw); the Research Institute for Diseases in the Elderly (RIDE); The Netherlands Genomics Initiative (NGI); the Ministry of Education, Culture and Science; the Ministry of Health, Welfare and Sports; the European Commission (DG XII); and the Municipality of Rotterdam. The UK Biobank was established by the Wellcome Trust medical charity, Medical Research Council, Department of Health, Scottish Government and the Northwest

0.26–0.80). Concerning rs1057910 (*CYP2C9\*3*), fall risk was increased in CA/variant allele carriers compared to AA/non-variant allele carriers (OR = 1.95, 95% CI 1.17–3.27). Regarding, rs1045642 (*ABCB1*), fall risk was increased in AG/variant allele carriers compared to GG/non-variant allele carriers (OR = 1.69, 95% CI 1.07–2.69). Concerning the *ABCB1*-haplotype (rs1045642/rs1128503), fall risk was increased in AA-AA/variant allele carriers compared to GG-GG/non-variant allele carriers (OR = 1.86, 95% CI 1.05–3.29). In the UK Biobank, in antidepressant users(n = 34,000) T/variant-allele of rs28371725 (*CYP2D\*41*) was associated with increased fall risk (OR = 1.06, 95% CI 1.01–1.12). G/non-variant-allele of rs4244285 (*CY2C19\*2*) was associated with decreased risk (OR = 0.96, 95% CI 0.92–1.00).

## Conclusion

This is the first study showing that certain genetic variants modify antidepressant-related fall risk. The results were not always consistent across the studies and should be validated in a study with a prospective design. However, pharmacogenetics might have value in antidepressant (de)prescribing in falls prevention.

## Introduction

Falls are a major public health problem in older adults. Annually, one-third of the people over the age of 65 years fall at least once a year and half of this group is at increased risk for recurrent falls [1]. Ten percent of fall incidents result in serious injuries such as fractures, head injuries, or even death [2, 3]. The high prevalence and injury rate lead to substantial healthcare costs. Fall injuries are among the 20 most expensive medical conditions among community-dwelling older adults [4]. Furthermore, older adults often experience a decline in functional status and social activities after a fall and report a reduced quality of life up to 9 months after the injury [3, 5]. Multiple intrinsic and extrinsic risk factors are associated with falling, with prescribed medications being one of the most important contributors [6, 7]. Antidepressants and their subclasses have consistently been associated with increased fall risk [8]. Antidepressant use may affect fall risk through several underlying pathways, among important cardiovascular and neurological effects [9, 10]. In general, falls can be considered a common adverse drug reaction (ADR) of antidepressant use.

Furthermore, ADRs are common in antidepressant users. The occurrence of ADRs other than falls and therapy effectiveness have been shown to be partly dependent on individual characteristics such as genetic variation [11–14]. Pharmacokinetic data underlines the significance of the cytochrome P450 (CYP) family in antidepressant metabolism. Especially, *CYP2D6* and *CYP2C19* are of major importance in the metabolization of many commonly prescribed antidepressants [11, 12, 15]. Functional genetic variants within these genes result in significant variation in enzymatic activity and subsequently, altered phenotypes (e.g. poor metabolism) and complementary pharmacokinetic parameters (e.g. higher drug concentration) [11, 14, 16]. Several other genes outside the CYP450 system have also been associated with antidepressants pharmacokinetics. For instance, the *ABCB1* gene encodes for P-glycoprotein (P-gp), a transmembrane protein located at the luminal membrane of the endothelial cells that form the blood-brain-barrier (BBB). At the BBB, P-gp acts as an active efflux pump and regulates intracerebral concentrations and may affect the clinical response of drugs that are

Regional Development Agency. It has also received funding from the Welsh Government, British Heart Foundation, Cancer Research UK and Diabetes UK. UK Biobank is supported by the National Health Service (NHS). UK Biobank is open to bona fide researchers anywhere in the world, including those funded by academia and industry. The medical research project is a non-profit charity which has received core funding of around £133 Million. Core funding continues to be received from the Wellcome Trust, the MRC, and more recently, from Cancer Research UK and NIHR. UK Biobank has received additional funding for: genotyping of all 500,000 participants (from the Department of Health, Medical Research Council, British Heart Foundation) The sponsors have no role in the design or implementation of the study, data collection, data management, data analysis, data interpretation, or in the preparation, review, or approval of the manuscript.

**Competing interests:** The authors have declared that no competing interests exist.

substrates for this transporter, such as antidepressants. Genetic variants within *ABCB1* have been associated with variation of treatment response of antidepressant therapy and conflicting results exist with respect to adverse events [17–19].

Different clinical prescribing guidelines for antidepressants exist (e.g. the Clinical Pharmacogenetics Implementation Consortium (CPIC) and the Dutch Pharmacogenetic Working Group (DPWG)), providing recommendations regarding dosing and drug choice based on predicted enzymatic activity [20–22]. Genotype based dose adjustments might be of value in reducing the risk of medication-related falls since the risk for falls has been shown to be dose-dependent [23–25]. In this case, falls prevention could be one of the target areas for pharmacogenetic testing. Therefore, our aim was to assess if candidate polymorphisms are associated with fall risk in antidepressant users.

## Methods

### Identification of genetic polymorphisms

The Pharmacogenomics Knowledgebase was used to identify relevant single nucleotide polymorphisms (SNPs) for the pharmacokinetics and adverse effects of antidepressants [26]. For supporting literature, a structured explorative search in PubMed was conducted by the first author (June 2019). The MESH-terms "antidepressants agents", "selective serotonin reuptake inhibitors", "tricyclic antidepressive agents", "serotonin and noradrenaline reuptake inhibitors", "pharmacogenetics" or "single nucleotide polymorphism" and "pharmacokinetics" or "adverse events" were combined in the search. We searched for clinically relevant SNPs for commonly prescribed antidepressants: paroxetine, (es)citalopram, sertraline, fluoxetine, amitriptyline, nortriptyline, clomipramine, imipramine, duloxetine, mirtazapine and venlafaxine. The following non-rare variants (minor allele frequency (MAF) >1% in Caucasian population) that could be genotyped from genome-wide arrays were identified [27, 28]: CYP1A2: rs762551 (*1F); CYP3A4: rs35599367 (*22); CYP3A5: rs776746 (*3); CYP2C9:rs1799853 (*2); rs1057910 (*3); CYP2D6: rs28371725 (*41); rs3892097 (*4); CYP2C19: rs4244285 (*2), rs12248560 (*17); ABCB1: rs1045642, rs1128503. Chromosome positions for these different SNPs are shown in S1 Table.

### Study population

Firstly, we investigated whether candidate genetic polymorphisms modify the association between antidepressant use and fall risk in a harmonized cohort of Dutch studies. Subsequently, we assessed if these polymorphisms were associated with fall risk in an existing genome-wide association study (GWAS) for drug-related falls conducted in UK Biobank (UKBB) Study.

For the candidate gene analysis, data from an existing cohort created for AD*F*ICE_IT study was used [29]. This is a harmonized cohort of five European cohorts (the Longitudinal Aging Study Amsterdam (LASA), the B-vitamins for the Prevention Of Osteoporotic Fractures (B-PROOF), the Rotterdam Study (or Erasmus Rotterdam Gezondheid Onderzoek (ERGO), the Irish Longitudinal Study on Ageing and the Activity and Function in the Elderly in Ulm study). Since only the LASA, B-PROOF and ERGO cohort have genetic data available, this subset was included in the current study. More detailed description of these cohorts can be found in S1 Appendix. Subsequently, we leveraged data from a genome-wide association study (GWAS) for drug-related falls performed in UKBB.

All studies were approved by medical ethics committees. For the B-PROOF study the Medical Ethics Committee (METC) of Wageningen University approved the study protocol [30]. The LASA study received approval by the METC of the VU University Medical Center [31].

The Rotterdam study received approval by the METC of the Erasmus MC and the Dutch Ministry of Health, Welfare and Sport [32]. The UKBB has approval from the North West Multi-centre research ethics committee (MREC) as a research Tissue Bank (RTB) approval. This approval means that researchers do not require separate ethical clearance and can operate under the RTB approval [33, 34]. In all studies, all participants gave written informed consent before entering the study. All experimental studies were conducted according to principles in the Declaration of Helsinki.

## Falls

Fall occurrence during the past 12 months was used as outcome measure. In the harmonized cohort, the occurrence of falls in the past 12 months was based on retrospective reporting by questionnaires in person or through a telephone contact. In the UKBB, self-reported data about falls was gathered through a touch screen questionnaire.

## Antidepressant use

Anatomical Therapeutic Chemical (ATC) codes were used to define antidepressant use in all cohorts. Our exposure of interest was antidepressant use (N06A and N06CA). In addition, we defined non-selective monoamine reuptake inhibitor (N06AA, N06CA01, and N06CA02), selective serotonin reuptake inhibitor (N06AB and N06CA03) and other antidepressant (N06AF, N06AG and N06AX) use. N06AF, N06AG categories were combined with N06AX (into category: 'other antidepressants') as there needed to be sufficient amount of users per variable. The antidepressant variables needed to be used by >1% of the participants.

In the LASA cohort, use of medication was assessed from questionnaires and checked with medication boxes provided by participants [35]. In B-PROOF, self-reported medication use was recorded by questionnaires at baseline [30]. In the Rotterdam study, medication use was registered during the home interview by participants presenting all the medication they were using at the day of the interview [36].

In the UKBB, self-reported regular medication and health supplements taken weekly, monthly or every three months were recorded at baseline [37]. Medication and health supplements data (Data field:20003) were manually mapped to identify antidepressant users.

## Covariates

In the harmonized cohort, characteristics including age, gender, living situation, educational level, pain, smoking habits and alcohol use were assessed using a questionnaire. During study visits, various characteristics were measured, including weight, height, blood pressure, depressive symptoms, anxiety symptoms and cognitive status (Mini-mental state examination (MMSE)). Additionally, serum creatinine levels were determined. Hand grip strength, balance and gait speed were assessed as measures of physical function. A detailed description of the covariate assessment in the harmonized cohort and the harmonization procedure is presented in S2 Appendix & S2 Table. Possible mediators between antidepressant use and falls (dizziness, balance, hypotension, gait speed) were not tested as covariates.

## Genotyping

Genotyping for the first LASA cohort (participants at the C wave) was performed by two arrays Axiom-NL Array and the Infinium Global Screening Array (GSA) [38]. The latter was also used for the genotyping of participants in 3B wave. Furthermore, Illumina-Omni express array was used in B-PROOF, and genotyping in the Rotterdam Study, was carried out with the

Illumina 550, 550 duo and Human 610 Quad arrays [30, 39]. After initial genotyping, stringent quality control was performed in all cohorts. The imputation data was based on the Haplotype Reference Consortium (HRC) panel version 1.1 in these cohorts, and the process was performed on the Michigan Imputation server. The imputation quality was good with an $R^2$ value higher than 0.88, 0.87 and 0.81 for the B-PROOF, LASA and Rotterdam Study, respectively. To extract the hard genotype calls (0,1 or 2) of relevant SNPs, PLINK version 1.90 was used [40]. In addition, the variants within *CYP2D6*, *ABCB1* and *CYP2C9* were combined into composed metabolism phenotype (S3 Table).

The majority of UKBB participants were genotyped with the Affymetrix UK Biobank Axiom Array (Santa Clara, CA, USA), while 10% of participants were genotyped with the Affymetrix UK BiLEVE Axiom Array. Detailed quality control and imputation procedures are described elsewhere [41]. Imputation was performed using the HRC Panel. When a SNP was not present, then the UK10K and 1000 Genomes phase 3 reference panels were used [42, 43].

## Statistical analysis

In the harmonized cohort, baseline characteristics were determined for antidepressant users and non-antidepressant users. Differences between groups were tested using a t-test for normally distributed variables, Mann-Whitney U test for non-normally distributed continuous variables and a Chi-square test for categorical variables. Deviation of allele frequencies from Hardy-Weinberg equilibrium (HWE) was assessed using a Chi-Square test.

To investigate the association between antidepressant use (and subgroups of antidepressants) and falls, logistic regression models were conducted. Initially, the models were adjusted for age and gender (model 1). Additionally, a second model (model 2) was examined assessing the following variables as potential covariates: cohort index, age, gender, education, body-mass index, alcohol use, smoking habits, use of a walking aid, living situation, MMSE, depressive symptoms, anxiety symptoms, diabetes, grip strength, co-medication, pain and kidney function. The covariates were examined separately and included in the model if they changed the Odds Ratio (OR) of the association between antidepressant use and falls by at least 10%.

To account for missing covariates, missing data was imputed in five copies with imputations by chained equation and the parameter estimates were pooled using Rubin's rules [44]. Among the 27 variables that were included in this analysis, there were 13 variables without missing values. The median of the proportion of missing values for the 14 variables with missing values was 8.9% (interquartile range, 0.6%-13.1%). We also tested the correlation between antidepressant use and the variables included in model 2 by Spearman's correlation coefficient. Various cut-offs (0.4–0.85) have been applied in the literature [45]. We applied the more restrictive cut-off of $\geq$0.4 for the coefficient to be sure that the multicollinearity problem is not present in our analyses.

To test for effect modification, an interaction term between the medication group and the SNP or composed metabolism phenotype was added to the age- and gender adjusted model (model 1). The exposure variable was first restrained to antidepressants relevant to the specific SNP (substrate specific) or metabolism phenotype according to the literature. Subsequently, the exposure variable was expanded to all antidepressants to create more power. In case of a p-value of the interaction term <0.1, further analyses for this SNP were conducted. The association between genotype and fall risk was assessed in antidepressant users. In addition, fall risk for antidepressants users was compared to non-users, stratified for different genotype carriers based on the individual SNPs and composed metabolism phenotypes.

Secondly, candidate gene analyses were performed in data from a GWAS for drug-related falls performed in UKBB. Users of 1) all antidepressants, 2) Tricyclic antidepressants (TCAs),

3) Selective Serotonin Reuptake Inhibitors (SSRIs) were selected and genetic association analyses for falls were performed in these subgroups. Logistic regression models were used and adjusted for age, gender and the first 15 principal components. Associations between SNPs and fall risk were tested for all autosomal variants. Individuals were excluded based on unusually high heterozygosity or >5% missing genotype rate and on a mismatch between self-reported and genetically-inferred sex. SNP exclusions were made based on low MAF (<1%), HWE (p >1 x 10$^{-06}$) and SNP call rate (>95%). We investigated the results of these analyses for the candidate SNPs mentioned above. P-values less than 0.05 were considered statistically significant.

Statistical analyses were conducted in SPSS (version 25.0, IBM, Armonk, NY, USA) and in R (version 3.6.1) using packages mice and miceadds.

## Results

### Harmonized cohort

**Study population.** Fall and medication data was available for 11,765 participants. Of them, 26.6% reported a fall in the past 12 months. In total, for 9,335 participants genetic data was available. In this group, 26.5% of the participants experienced a fall in the past 12 months. The baseline characteristics of antidepressant users and non-antidepressant users are shown in S4 Table. In summary, antidepressant users were more often female, more often smokers, experienced more anxiety and depressive symptoms and reported more often dizziness and pain. Antidepressant users had poorer physical function parameters and experienced more often hypotension compared to non-antidepressant-users. Also, antidepressant users used more medication including more often benzodiazepines and opioids.

**Antidepressant use and fall risk.** Of the 11,765 participants, 700 participants (5.9%) were antidepressant users. Antidepressant use was associated with increased fall risk in the fully adjusted model (OR 1.58, 95% CI 1.34–1.87) and similarly to subclasses of antidepressants. The results are shown in Table 1. There were no signs of significant multicollinearity between the variables included in model 2 and antidepressant use.

**Genotypes and antidepressant-related fall risk.** Allele frequencies of the analyzed genotypes and the chi-square tests of the distribution according to HWE are presented in S5 Table. Allele frequencies of rs1799853 (*CYP2C9*2*) deviated significantly from HWE. Sensitivity analyses in each cohort showed a significant deviation only in the LASA C Cohort (S6 Table).

**Table 1. Association between antidepressant use and fall occurrence in past 12 months.**

|  | N | Model 1 (OR 95% CI) [a] | Model 2 (OR 95% CI) [b] |
|---|---|---|---|
| **Antidepressant use** | 700 | 1.81 (1.54–2.13)* | 1.58 (1.34–1.87)* |
| **Classified by type of antidepressant** |  |  |  |
| SSRI | 376 | 1.86 (1.77–2.19)* | 1.53 (1.23–1.90)* |
| TCA | 199 | 1.72 (1.29–2.31)* | 1.43 (1.07–1.92)* |
| Other Antidepressants | 138 | 1.88 (1.33–2.67)* | 1.51 (1.06–2.15) * |

Data is presented in odds ratio and 95% confidence interval; N = Number of antidepressant users from 11,765 participants included in analysis. SSRI = Selective Serotonin Reuptake Inhibitor; TCA = Tricyclic Antidepressants. a Model 1 was adjusted for age and gender. b Model 2 of antidepressant and SSRI use was adjusted for age, gender and depressive symptoms. Model 2 of TCA and other antidepressant use was adjusted for age, gender, depressive symptoms and number of used medications. * statistically significant at p<0.05.

When the exposure term was limited to antidepressants being relevant for the metabolism of the specific SNP, interaction term including rs28371725 (*CYP2D6*41*) was significant. When the exposure term was 'all antidepressants', significant interaction terms were found for the following SNPs: rs28371725 *(CYP2D6*41)*, rs28371725/rs3892097 *(CYP2D6*41/CYP2D6*4)* combined into *CYP2D6* phenotype, rs1057910 *(CYP2C9*3)*, rs1045642 *(ABCB1)* and rs1045642/rs1128503 combined into *ABCB1* haplotype.

The results of the associations between *CYP2D6* genotype and fall risk within antidepressant users are presented in Table 2. In antidepressant users, fall risk was decreased in the heterozygote TC (variant) allele carriers of rs28371725 (*41) compared to CC (non-variant) allele carriers (OR 0.45, 95% CI 0.26–0.80). Also, carriers of at least one variant allele (TC and TT) of rs28371725 had significant decreased fall risk compared to non-variant allele (CC) carriers (OR 0.47, 95% CI 0.27–0.80). When *4 and *41 were combined into *CYP2D6* phenotypes, no significant associations were found. The antidepressant users in different CYP2D6*41 genotype carriers and for the CYP2D6 phenotype genotype carriers did not differ in terms of Fall risk increasing drug (FRID) use, CYP2D6 inhibitor or CYP2D6 inducer use.

The results of the associations between antidepressant use and fall risk stratified for the genotypes of *CYP2D6* are presented in S7 Table. Antidepressant use was associated with increased fall risk in the non-variant allele (CC) carriers of rs28371725 (*41) (OR 1.86, 95% CI 1.53–2.26). No significant association was found in the category of at least one variant allele carriers (TC and TT). When *4 and *41 were combined into phenotypes, antidepressant use was associated with increased fall risk in the extensive metabolizer (EM) (OR 1.73, 95% CI 1.43–2.09) and in the poor metabolizer (PM) phenotype (OR 2.17, 95% CI 1.02–4.61).

The results of the associations between *CYP2C9* genotype and fall risk within antidepressant users are presented in Table 3. An increased fall risk was found in the heterozygote allele (CA) carriers of rs1057910 compared to non-variant allele carriers (AA) (OR 1.95, 95% CI

**Table 2. Association between CYP2D6*41 and CYP2D6*41/*4 genotype and fall risk within antidepressant users.**

| | | All antidepressant users | | | Substrate specific antidepressant users [#] | |
|---|---|---|---|---|---|---|
| **CYP2D6 *41** | N | Model 1 | P-value | N | Model 1 | P-value |
| CC | 484 | Ref | | 304 | Ref | |
| TC | 75 | 0.45 (0.26–0.80) | 0.006* | 49 | 0.47 (0.23–0.94) | 0.034* |
| TT | 7 | 0.61 (0.12–3.22) | 0.559 | 4 | - | - |
| | | | | | | |
| CC | 484 | Ref. | | 304 | Ref. | |
| Variant allele carriers (TC and TT) | 82 | 0.47 (0.27–0.80) | 0.006* | 53 | 0.43 (0.21–0.85) | 0.016* |
| **CYP2D6 phenotype** | N [#] | Model 1 | P-value | N [#] | Model 1 | P-value |
| EM | 516 | Ref | | 326 | Ref | |
| IM | 19 | 0.30 (0.08–1.04) | 0.058 | 12 | 0.32 (0.07–1.50) | 0.147 |
| PM | 31 | 1.38 (0.66–2.90) | 0.398 | 19 | 1.39 (0.54–3.59) | 0.494 |
| | | | | | | |
| EM | 516 | Ref | | 326 | Ref. | |
| IM and PM | 50 | 0.84 (0.45–1.57) | 0.589 | 31 | 0.86 (0.39–1.88) | 0.704 |

Data is presented in odds ratio and 95% confidence interval.

N = number of participants per genotype. EM = Extensive metabolizer; IM = Intermediate metabolizer; PM = Poor metabolizer; Model 1 was adjusted for age and gender.

[#] users of the following antidepressants were included in the analysis: *amitriptyline, nortriptyline, clomipramine, imipramine, paroxetine, fluvoxamine, citalopram, sertraline, doxepin, duloxetine, mirtazapine, venlafaxine, trazodone.*

*statistically significant at p<0.05.

**Table 3. Association between CYP2C9\*3 and fall risk within antidepressant users.**

| CYP2C9 *3 | All antidepressant users | | | Substrate specific antidepressant users # | | |
|---|---|---|---|---|---|---|
| | N | Model 1 | P-value | N | Model 1 | P-value |
| AA | 496 | Ref | | 113 | Ref | |
| CA | 70 | 1.95 (1.17–3.27) | 0.011* | 17 | 1.79 (0.62–5.16) | 0.283 |
| CC | - | - | | - | | |
| Any variant allele carriers (CA & CC) | 70 | 1.95 (1.17–3.27) | 0.011* | 17 | 1.79 (0.62–5.16) | 0.283 |

Data is presented in odds ratio and 95% confidence interval.

N = number of participants per genotype Model 1 was adjusted for age and gender.

# users of the following antidepressants were included in the analysis: *fluoxetine (and psycholeptics),escitalopram, amitriptyline (and psycholeptics).*

*statistically significant at p<0.05.

1.17–3.27). The antidepressant users in different CYP2C9*3 genotype carriers did not differ in terms of FRIDs use, CYP2C9 inhibitor or CYP2C9 inducer use.

The results of the associations between antidepressant use and fall risk stratified for the genotypes of rs1057910 of the *CYP2C9* gene are presented in S8 Table. Antidepressant use was associated with increased fall risk in the non-variant allele (AA) carriers (OR 1.56, 95% CI 1.28–1.89), in the heterozygote (CA) allele carriers (OR 2.98, 95% CI 1.82–4.88) and in the carriers of at least one variant allele (CA or CC) (OR 3.02, 95% CI 1.84–4.93).

The results of the associations between rs1045642 genotype and rs1045642/rs1128503 haplotype genotype and fall risk in antidepressant users are presented in Table 4. An increased fall risk was found in the heterozygote variant allele (AG)-carriers compared to non-variant (GG) allele carriers (OR 1.69, 95% CI 1.07–2.69) as well as in GA and AA variant allele carriers compared to non-variant allele (GG) carriers (OR 1.65, 95% CI 1.06–2.55). Regarding the haplotype, variant allele (AAAA) carriers had an increased fall risk compared to non-variant allele (GGGG) carriers (OR 1.86, 95% CI 1.05–3.29) as well as the combined variant allele (AAAA or GAGA)-carriers (OR 1.72, 95% CI 1.07–2.77). The antidepressant users in different ABCB1 genotype carriers did not differ in terms of FRIDs use, P-gp inhibitor or P-gp inducer use.

The results of the associations between antidepressant use and fall risk stratified for the genotypes of rs1045642 and rs1045642/rs1128503 haplotype of the *ABCB1* gene are presented in S9 Table. Antidepressant use was associated with an increased fall risk in the heterozygote

**Table 4. Association between ABCB1 rs1045642 and rs1045642/rs1128503 haplotype and fall risk within antidepressant users.**

| rs1045642 | All antidepressant users | | |
|---|---|---|---|
| | N | Model 1 | P-value |
| GG | 127 | Ref. | |
| GA | 271 | 1.69 (1.07–2.69) | 0.025* |
| AA | 168 | 1.57 (0.95–2.60) | 0.080 |
| Variant allele carriers (GA and AA) | 439 | 1.65 (1.06–2.55) | 0.025* |
| **rs1045642/rs11285003** | | | |
| GGGG | 115 | Ref. | |
| GAGA | 190 | 1.65 (0.99–2.74) | 0.054 |
| AAAA | 103 | 1.86 (1.05–3.29) | 0.034* |
| Variant allele carriers (GAGA & AAAA) | 293 | 1.72 (1.07–2.77) | 0.026* |

Data is presented in odds ratio and 95% confidence interval. N = number of participants per genotype. Model 1 was adjusted for age and gender.

*statistically significant at p<0.05.

**Table 5. Association between candidate SNPs and fall risk in antidepressant users in UKBB.**

| Gene | SNP | EA[a] | NEA[b] | EAF[c] | OR (95% CI) | P-value |
|------|-----|-------|--------|--------|-------------|---------|
| ABCB1 | rs1045642 | A | G | 0.54 | 1.01 (0.98–1.05) | 0.448 |
| | rs1128503 | A | G | 0.44 | 1.01 (0.98–1.05) | 0.443 |
| CYP3A4 | rs35599367 (*22) | G | A | 0.95 | 1.01 (0.94–1.09) | 0.761 |
| CYP3A5 | rs776746 (*3) | C | T | 0.93 | 1.00 (0.94–1.07) | 0.945 |
| CYP2C9 | rs1057910 (*3) | A | C | 0.94 | 0.99 (0.93–1.06) | 0.854 |
| | rs1799853 (*2) | C | T | 0.86 | 1.02 (0.97–1.07) | 0.416 |
| CYP2C19 | rs4244285(*2) | G [d] | A [e] | 0.85 | 0.96 (0.92–1.00) | 0.049* |
| | rs12248560 (*17) | C | T | 0.79 | 1.00 (0.96–1.04) | 0.982 |
| CYP1A2 | rs762551(*1F) | C | A | 0.27 | NA | NA |
| CYP2D6 | rs28371725(*41) | T [e] | C [d] | 0.10 | 1.06 (1.01–1.12) | 0.026* |
| | rs3892097 (*4) | T | C | 0.20 | 0.99 (0.95–1.03) | 0.496 |

[a] EA = Effect allele

[b] NE = Non-effect allele

[c] EAF = Effect allele frequency

[d] normal function allele

[e] deficient function allele.

* statistically significant at p<0.05.

variant allele (AG)-carriers (OR 1.92, 95% CI 1.48–2.48), in the homozygote variant allele (AA)-carriers (OR 1.83, 95% CI 1.31–2.54) and in AG or AA–carriers (OR 1.89, 95% CI 1.54–2.31). Regarding the haplotype, antidepressant use was associated with an increased fall risk in the heterozygote variant allele (GAGA)-carriers (OR 1.79, 95% CI 1.32–2.44), the homozygote variant allele (AAAA)-carriers (OR 2.19, 95% CI 1.44–3.33) and in the combined variant allele carriers (GAGA and AAAA) group (OR 1.92, 95% CI 1.50–2.46).

## UK Biobank

UKBB GWAS for drug-related falls included data from approximately 34,000 antidepressant users, almost 12,000 TCA users, and approximately 12,000 SSRI users. Around 35% of the antidepressant users reported at least one fall in the past year. Table 5 shows the results of the association analysis for falls in antidepressant users. In antidepressant uses, T (variant) allele of rs28371725 located in *CYP2D6* gene (*41) was associated with increased risk of falling (OR 1.06, 95% CI 1.01–1.12). Furthermore, the G (non-variant) allele of rs4244285 located in *CYP2C19* gene (*2) was associated with decreased risk of falling (OR 0.96, 95% CI 0.92–1.00). S10 Table shows the results of the association analysis for SSRI users. In SSRI users, C (non-variant)-allele of rs1799853 located in *CYP2C9* gene (*2) was associated with increased risk of falling (OR 1.08, 95% CI 1.01–1.15). S11 Table provides the results for the TCA users. In TCA users, T (variant) allele of rs28371725 located in *CYP2D6* gene (*41) was associated with increased risk of falling (OR 1.12, 95% CI 1.02–1.22).

## Discussion

We addressed the effect of genetic variation on antidepressant-related falls and thus the potential for applying pharmacogenetics in this clinically very relevant health problem. In line with our expectations, antidepressant use was associated with increased fall risk, and the association was significantly modified by SNPs mapping to *CYP2D6*, *CYP2C9*, *CYP2C19* and *ABCB1*. However, the findings were not always consistent across the studies.

To our knowledge, this is the first study that investigated the influence of pharmacogenetics in relation to antidepressant-related fall risk. We assessed this topic in two study populations, in the UKBB with a very big sample size of middle-aged to older antidepressant users and in the smaller harmonized cohort of Dutch studies of older participants, with thus an overall higher risk of antidepressant-related falls given the older age. To date, data on pharmacokinetic differences is scarce especially for the population of older adults.

For the variant alleles *4 and *41 of the *CYP2D6* gene, we hypothesized an increased fall risk due to absent or diminished antidepressant metabolism. *4 is the most common variant allele of *CYP2D6* with a MAF of 15.5% in Caucasians and the most frequent non-functional allele in poor metabolizers (PMs) [27, 46]. Due to absent *CYP2D6*-mediated metabolism, PMs have higher antidepressant plasma concentrations than EMs (normal enzymatic function) and are therefore more likely to suffer from dose-dependent ADRs [47, 48]. *41 is the most common deficient allele in Caucasians, being present among 3% and leading to decreased enzyme activity (creating the IM phenotype) [27]. Results from the UKBB population are partly in line with our hypothesis, since the *41 variant allele was associated with an increased fall risk (OR 1.06) but concerning *4 variant allele, no significant association was found. In the harmonized cohort for the *4 variant allele no association was found and in antidepressant users, fall risk was decreased in the heterozygote carriers of *41 compared to non-variant carriers. When *4 and *41 were combined into *CYP2D6* phenotype in the harmonized cohort, no significant associations were found between genotype and fall risk in antidepressant users. However, the number of participants in the IM and PM groups was low. These contradictory findings regarding *41 could be related to several factors. First, *CYP2D6* plays an important role in the metabolism of TCAs and the proportion of TCA users was higher in UKBB than in the harmonized cohort. Second, in the harmonized cohort the number of variant allele carriers was limited and this could have played a role in the conflicting results. Third, with increasing age, a possible decline in CYP-enzyme activity is described and the enzymatic differences due to genotype differences may become smaller with age. However, others state that there are no important age-dependent changes in CYP-enzyme activity (especially in *CYP2D6* activity) [24, 49].

Concerning *CYP2C19* *2 variant allele, we hypothesized an increased fall risk due to diminished/absent antidepressant metabolism in carriers of *2 variant. This was indeed observed in the UKBB. Since not-carrying *2 variant was associated with decreased fall risk, so carrying the *2 variant was associated with increased fall risk. *2 variant allele is the most common non-function allele in CYP2C19, with a MAF of 18.3% in the Caucasian population [20, 22, 27]. (Es)citalopram is being primarily metabolized by *CYP2C19* and differences in pharmacokinetic properties have been reported based on phenotype [50]. Furthermore, Fabbri et al. showed that PMs had a higher risk of adverse effects early in treatment, but also had better symptom improvement and remission probability than EMs during (es)citalopram treatment [51]. Moreover, *CYP2C19* is involved in metabolizing the tertiary amines of TCAs (e.g. amitriptyline) to secondary amines (e.g.nortriptyline) and patients may have more risk of treatment failure or adverse effects based on their genotype [21]. In our harmonized cohort, no association with fall risk was found for this variant allele, which is probably related to the small cohort size. Considering the modest effect found in UKBB, we do not expect there is sufficient power present in the harmonized cohort.

Furthermore, in UKBB among SSRI users, not-carrying *2 allele of *CYP2C9* was associated with increased fall risk, so carrying *2 allele is associated with decreased fall risk. *3 allele variant showed no association in UKBB. In our harmonized cohort, no significant association was found for *2 and significantly increased fall risk was found in the heterozygote and variant allele carriers of *3 compared to normal carriers. Both *2 and *3 variant alleles are the most

common variants of *CYP2C9* and result in decreased or absent enzymatic activity. In Caucasians, *2 and *3 have MAF of about 12 and 8%, but result in poor metabolizer phenotype only in about 0.7% of Caucasians [15, 22, 52]. *CYP2C9* has been less studied in the field of antidepressant pharmacogenetics, since it does not play a primary role in the metabolism of antidepressants, but is involved in the secondary pathways [12]. *CYP2C9* forms the only secondary pathway of metabolism of fluoxetine, and therefore decreased enzymatic activity of *CYP2C9* could be of importance in co-existence with *CYP2D6* PM phenotype in fluoxetine users [12, 52]. *CYP2C9* genotypes should probably be investigated in conjunction with other metabolizing enzymes and considering the limited evidence related to the role of *CYP2C9* in antidepressant metabolism our results need to be replicated.

*ABCB1* is another candidate gene that might modulate antidepressant response [49]. In the harmonized cohort, in antidepressant users, fall risk was increased in the heterozygote (AG) genotype and variant allele carriers of the rs1045642 SNP compared to the GG (non-variant) genotype carriers. In the UKBB population, no association with fall risk was found. Polymorphisms of the *ABCB1* gene appear to be correlated to expression levels and function of P-gp. These SNPs have been associated with changes in pharmacokinetics of substrate drugs, treatment response and adverse effects [53]. However, most studies that analyzed the relation of *ABCB1* gene variants with antidepressants have focused on treatment response and show contradictory results [18, 19, 54–56]. A significant association between rs1045642 and the risk of postural hypotension in nortriptyline users has been shown [18]. Noteworthy is the importance of postural hypotension as a fall risk factor. Furthermore, the *ABCB1* gene is large and highly polymorphic. Rs1045642(3435C>T) shows strong linkage disequilibrium with rs1128503(1236C>T) and rs2032582(2677G>T/A). The rs1128503 (1236C)—rs2032582 (2677G)—rs1045642 (3435C) (C-G-C) haplotype is frequently observed in most populations [53]. This haplotype has been linked to altered P-gp function in several studies, but also these reports have been inconclusive and most found no correlation with therapeutic response [53, 54, 57]. Since rs2032582 was not available for analysis in our study (given this SNP is tri-allelic), we analyzed the combination of rs1045642 and rs1128503. In antidepressant users, we found an increased fall risk in the homozygote variant allele carriers (AA-AA) compared to the non-variant (GG-GG) genotype carriers. Although it is difficult to draw conclusions from this, due to the incomplete haplotype, it could be a first sign that determining the full haplotype could be relevant in antidepressant-related fall risk.

Our study has several limitations. First, in the harmonized cohort despite the large number of participants with genetic data the number of participants in the different genotypes was relatively low due to relatively uncommon genetic variants. Second, is the cross-sectional character of the study. Although cross-sectional studies are usually vulnerable to recall bias, no prospective data on falls was available for all included cohorts. Third, in this cross-sectional study ascertainment of antidepressant use was done at study visits, not at time of the fall. Thus, antidepressant usage during the fall incident might have differed. This may have led to misclassification and as such dilution of the effect. Therefore, it is necessary to validate our findings in a study with a prospective study design. Fourth, because we analyzed prevalent use of antidepressants, confounding by contraindication or depletion of susceptibles might have biased the associations towards the null. Fifth, as the majority of the study population was of Caucasian ethnicity, our results cannot be extrapolated to other ethnic groups. Sixth, for both *CYP2D6* and *CYP2C19* no data was available on the duplicate active alleles leading to the ultrarapid metabolizer phenotype, as duplication of genes could not be detected by the genome arrays used in our cohorts. However, duplication of CYP2D6 occur in about 1–2 percent in Caucasian Dutch [27, 58]. Seventh, CYP2D6 is highly polymorphic with over 100 known allelic variants [59], however, we were limited to two of the known variants. Eight, multiple metabolic

pathways are involved in the metabolism of most drugs and for example combinatorial phenotypes based on both *CYP2D6* and *CYP2C19* genes, should be assessed in future studies. Eight, different polymorphisms can affect the pharmacokinetics and pharmacodynamics of each antidepressant differently and this distinction could not be made in our analyses.

### Clinical implications and future perspectives

There is an increasing need to better personalize antidepressant therapy to optimize treatment effects and prevent adverse events like falling. Our current study has no direct clinical implications, since our findings are novel and need to be validated with a prospective design. In general, clinical guidelines have already been developed for different *CYP2D6* and *CYP2C19* phenotypes, and recommendations are provided regarding dosing of the antidepressant and drug choice. For example, for amitriptyline and nortriptyline users a 50% dose reduction is strongly advised in case of *CYP2D6* and *CYP2C19* PM phenotype [20, 21]. Concerning future research, investigating the pharmacogenomics effects during the first few weeks after initiating antidepressant therapy would be of interest, as fall risk appears to be highest in this time frame [60]. Furthermore, studies on blood concentration levels in relation to fall risk are needed to confirm our hypothesized mechanism.

### Conclusion

This is the first study to show that genetic variation appears to modify the association between antidepressant use and fall risk. Polymorphisms in *CYP2D6*, *CYP2C19*, *CYP2C9* and *ABCB1* modified the association between antidepressant use and fall risk, although the findings were not consistent across the studies. Before firm conclusions can be made, future research with a prospective design is needed. Nonetheless, our results are a first sign that possibly implementation of pharmacogenetic applications in antidepressant (de)prescribing in falls prevention will be of clinical value in the future.

### Supporting information

**S1 Appendix. Cohort description.**
(DOCX)

**S2 Appendix. Covariate assessment.**
(DOCX)

**S1 Table. Chromosome positions.** Chromosome position is based on assembly GRCh37.p13. $R^2$ allele correlation was calculated with a help of LDpair tool (LDlink | An Interactive Web Tool for Exploring Linkage Disequilibrium in Population Groups (nih.gov)).
(DOCX)

**S2 Table. Details regarding covariate assessment in different cohorts and harmonization algorithms.**
(DOCX)

**S3 Table. Phenotypes of SNPs available in harmonized dataset.** WT = wild type; HeZ = Heterozygous for variant allele; HoZ = Homozygous for variant allele; EM = Extensive Metabolizer; IM = Intermediate Metabolizer; PM = Poor Metabolizer.
(DOCX)

**S4 Table. Baseline characteristics of antidepressant users and non-antidepressant users.** [a] mean (SD), [b] presented as n (%), [c] presented as median (IQR), [d] mean z-scores (SD), [e] data

available in LASA and ERGO-5; HADS = Hospital Anxiety and Depression scale; MMSE = Mini-Mental State Examination; GFR = Glomerular filtration rate according to Cockcroft and Gault formula. * statistically significant at p-value <0.05.
(DOCX)

**S5 Table. Allele frequencies and distribution test according to Hardy-Weinberg equilibrium.** Analysis of cases with complete medication, fall and genetic data. Values are presented as N. Deviation of allele frequencies from Hardy-Weinberg equilibrium was assessed using a Chi-Square test. * If p<0.05, SNP is not consistent with Hardy Weinberg Equilibrium.
(DOCX)

**S6 Table. Allele frequencies of rs1799853 and distribution test according to Hardy-Weinberg equilibrium in each cohort.** Analysis of cases with complete medication, fall and genetic data. Values are presented as N. Deviation of allele frequencies from Hardy-Weinberg equilibrium was assessed using a Chi-Square test. * If p<0.05, SNP is not consistent with Hardy Weinberg Equilibrium.
(DOCX)

**S7 Table. Association between antidepressant use and fall risk, stratified for CYP2D6*41 genotype and CYP2D*41/*4 phenotype.** Data is presented in odds ratio and 95% confidence interval. Model 1 was adjusted for age and gender. N = number of participants per genotype (total includes also participants not using antidepressants). EM = Extensive metabolizer; IM = Intermediate metabolizer; PM = Poor metabolizer. # rs28371725 (*41) & rs3892097 (*4) combined. $ users of the following antidepressants were included in the exposed category: amitriptyline, nortriptyline, clomipramine, imipramine, paroxetine, fluvoxamine, citalopram, sertraline, doxepin, duloxetine, mirtazapine, venlafaxine, trazodone. * statistically significant at p<0.05.
(DOCX)

**S8 Table. Association between antidepressant use and fall risk, stratified for CYP2C9*3 genotype.** Data is presented in odds ratio and 95% confidence interval. Model 1 was adjusted for age and gender. N = number of participants per genotype (total includes also participants not using antidepressants). # users of the following antidepressants were included in the exposed category: fluoxetine (and psycholeptics), escitalopram, amitriptyline (and psycholeptics). *statistically significant at p<0.05.
(DOCX)

**S9 Table. Association between antidepressant use and fall risk, stratified for ABCB1 (rs1045642) and rs1045642/rs1128503 haplotype.** Data is presented in odds ratio and 95% confidence interval. Model 1 was adjusted for age and gender. N = number of participants per genotype (total includes also participants not using antidepressants). *statistically significant at p<0.05.
(DOCX)

**S10 Table. Association between candidate SNPs and fall risk in SSRI users in UKBB.** [a] EA Effect allele, [b] NEA Non-effect allele, [c] EAF Effect allele frequency, [d] normal function allele, [e] deficient function allele. *statistically significant at p<0.05.
(DOCX)

**S11 Table. Association between candidate SNPs and fall risk in TCA users in UKBB.** [a] EA Effect allele, [b] NEA Non-effect allele, [c] EAF Effect allele frequency, [d] normal function allele, [e]

deficient function allele. *statistically significant at p<0.05.
(DOCX)

## Acknowledgments

**B-PROOF:** we thank the participants of the B-PROOF study for their enthusiasm and cooperation. Furthermore, we thank the dedicated research team that conducted the study.

**LASA:** we thank the participants of the LASA study. We are grateful too all the researchers and fieldworkers at LASA for their ongoing commitment to the study.

**Rotterdam study:** we thank the participants and staff from the Rotterdam study.

**UKBB:** We are very grateful to all UK Biobank participants and staff. This research has been conducted using data from UK Biobank, a major biomedical database (www.ukbiobank. ac.uk).

## Author Contributions

**Conceptualization:** A. C. Pronk, L. J. Seppala, N. van der Velde.

**Data curation:** A. C. Pronk, L. J. Seppala, K. Trajanoska, N. Stringa, B. van de Loo, L. C. P. G. M. de Groot, N. M. van Schoor, F. Koskeridis, G. Markozannes, E. Ntzani, A. G. Uitterlinden, F. Rivadeneira, B. H. Stricker, N. van der Velde.

**Formal analysis:** A. C. Pronk, L. J. Seppala, K. Trajanoska, B. van de Loo, F. Koskeridis, G. Markozannes.

**Supervision:** L. J. Seppala, N. van der Velde.

**Writing – original draft:** A. C. Pronk, L. J. Seppala, N. van der Velde.

**Writing – review & editing:** A. C. Pronk, L. J. Seppala, K. Trajanoska, N. Stringa, B. van de Loo, L. C. P. G. M. de Groot, N. M. van Schoor, F. Koskeridis, G. Markozannes, E. Ntzani, A. G. Uitterlinden, F. Rivadeneira, B. H. Stricker, N. van der Velde.

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
