## [Decision Letter · Decision Letter 0]

20 Jan 2022

PONE-D-21-37547Candidate Genetic variants and Antidepressant-related fall risk in middle-aged and older adultsPLOS ONE

Dear Dr. Pronk,

Thank you for submitting your manuscript to PLOS ONE. After careful consideration, we feel that it has merit but does not fully meet PLOS ONE’s publication criteria as it currently stands. Therefore, we invite you to submit a revised version of the manuscript that addresses the points raised during the review process.

We look forward to receiving your revised manuscript.

Kind regards,

Francesc Calafell

Academic Editor

PLOS ONE

Journal Requirements:

When you resubmit, please ensure that you provide the correct grant numbers for the awards you received for your study in the ‘Funding Information’ section."

This work was supported by the Clementine Brigitta Maria Dalderup fund (project numbers 20713 and 7303, receiver N. van der Velde), which is an Amsterdam University fund (https://www.auf.nl/en/about-the-fund/about.html). The funder had no role in study design, data collection and analysis, decision to publish, or preparation of the manuscript.

The initial B-PROOF study has received funding so far by the Netherlands Organization for Health Research and Development (ZonMw, Grant 6130.0031), The Hague; unrestricted grant from NZO (Dutch Dairy Association), Zoetermeer; Orthica, Almere; Netherlands Consortium Healthy Ageing (NCHA) Leiden/Rotterdam; Ministry of Economic Affairs, Agriculture and Innovation (project KB-15-004-003), The Hague; Wageningen University, Wageningen; VUmc, Amsterdam; Erasmus Medical Center, Rotterdam.

The Longitudinal Aging Study Amsterdam (LASA) is largely supported by a grant from the Netherlands Ministry of Health, Welfare and Sports, Directorate of Long-Term Care. The data collection in 2012–2013 was financially supported by the Netherlands Organization for Scientific Research (NWO) in the framework of the project “New Cohorts of young old in the 21st century” (File Number 480-10-014).

The Rotterdam Study is supported by the Erasmus MC University Medical Center and Erasmus University Rotterdam; The Netherlands Organisation for Scientific Research (NWO); The Netherlands Organisation for Health Research and Development (ZonMw); the Research Institute for Diseases in the Elderly (RIDE); The Netherlands Genomics Initiative (NGI); the Ministry of Education, Culture and Science; the Ministry of Health, Welfare and Sports; the European Commission (DG XII); and the Municipality of Rotterdam.

The UK Biobank was established by the Wellcome Trust medical charity, Medical Research Council, Department of Health, Scottish Government and the Northwest Regional Development Agency. It has also received funding from the Welsh Government, British Heart Foundation, Cancer Research UK and Diabetes UK. UK Biobank is supported by the National Health Service (NHS). UK Biobank is open to bona fide researchers anywhere in the world, including those funded by academia and industry. The medical research project is a non-profit charity which has received core funding of around £133 Million. Core funding continues to be received from the Wellcome Trust, the MRC, and more recently, from Cancer Research UK and NIHR. UK Biobank has received additional funding for: genotyping of all 500,000 participants (from the Department of Health, Medical Research Council, British Heart Foundation)

The sponsors have no role in the design or implementation of the study, data collection, data management, data analysis, data interpretation, or in the preparation, review, or approval of the manuscript.

Reviewers' comments:

Reviewer's Responses to Questions

**Comments to the Author**

1. Is the manuscript technically sound, and do the data support the conclusions?

Reviewer #1: Yes

2. Has the statistical analysis been performed appropriately and rigorously? 

Reviewer #1: I Don't Know

3. Have the authors made all data underlying the findings in their manuscript fully available?

Reviewer #1: No

4. Is the manuscript presented in an intelligible fashion and written in standard English?

Reviewer #1: Yes

5. Review Comments to the Author

Reviewer #1: This paper analyses the possible relation between some candidate pharmacogenes and falls in older antidepressant users.

Overall, the methodology, results, and conclusions seem sound.

Although I think that the paper is mostly suitable for publication, I do find, however, that it might also benefit from some improvements:

- gene CYP3A5 is mentioned in Table 5 and Suppl. T1, but is missing from the text.

- 6 cohorts are mentioned in the Methods, but only 5 are named (LASA, B-PROOF, ERGO, Irish LSA, AFE Ulm), and only 3 are explained in Suppl. 1 (ERGO, B-PROOF, LASA).

- falls are considered in the previous 12 months, but no time is considered for antidepressant use - I'm not sure whether this is accounted for in the limitations of the paper.

- it is not clear why some antidepressant categories have been analysed separately but others (monoamine oxidase inhibitors, i.e. N06AF and N06AG) are analysed together with "other antidepressants" (N06AX)

- it is not clear why some possible covariates were not analysed in Model 2 (e.g. dizzines, balance...)

- it is not clear why the cut-off of multicollinearity is set at 0.4 and not another value

- I am not sure whether the results of the supplementary tables regarding the associations between antidepressant use and fall risk stratified for genotypes are correct (e.g. Number of participants per genotype in group "All antidepressant users" being equal to group "Substrate specific antidepressant users", and generally being superior to the reported total number of Antidepressant users (N = 700)).

- it is not clear why the "alcohol use" covariate has been limited to groups with no use and high use

- some limitations of the study are not stated: trans people appear to be excluded from the analysis, all participants seem to have a BMI > 27 (overweight)

- terms "gender" and "sex" seem to be used interchangeably or not properly explained (e.g. "gender" in data table but "models adjusted by sex")

6. PLOS authors have the option to publish the peer review history of their article (what does this mean?). If published, this will include your full peer review and any attached files.

Reviewer #1: No

---

## [Author Response · Author response to Decision Letter 0]

11 Mar 2022

All points adressed by the reviewer and editor are listed in the rebuttal letter.

---

## [Editor Report · Decision Letter 1]

24 Mar 2022

Candidate genetic variants and antidepressant-related fall risk in middle-aged and older adults

PONE-D-21-37547R1

Dear Dr. Pronk,

We’re pleased to inform you that your manuscript has been judged scientifically suitable for publication and will be formally accepted for publication once it meets all outstanding technical requirements.

Kind regards,

Francesc Calafell

Academic Editor

PLOS ONE
---

## [Editor Report · Acceptance letter]

6 Apr 2022

PONE-D-21-37547R1 

Candidate genetic variants and antidepressant-related fall risk in middle-aged and older adults 

Dear Dr. Pronk:

I'm pleased to inform you that your manuscript has been deemed suitable for publication in PLOS ONE. Congratulations! Your manuscript is now with our production department. 

Kind regards, 

on behalf of

Dr. Francesc Calafell 

Academic Editor

PLOS ONE